# Comment on Rogers et al. The Combined Effects of Cannabis, Methamphetamine, and HIV on Neurocognition. *Viruses* 2023, *15*, 674

**DOI:** 10.3390/v15081753

**Published:** 2023-08-17

**Authors:** Patrick Anthony Augello, Jingwei Wu

**Affiliations:** College of Public Health, Temple University, Philadelphia, PA 19122, USA

We read with great interest the article by Rogers et al. (2023), *The Combined Effects of Cannabis, Methamphetamine, and HIV on Neurocognition*, published in *Viruses* [1]. The authors compared the influence of these substances, as well as HIV infection, on neurocognitive performance across multiple domains. We recognize the importance of this research in the understanding of neurocognitive decline, especially as it pertains to this vulnerable population. This research is imperative to increasing positive patient outcomes, including adherence to antiretroviral therapy. For this reason, we believe that it is important to inquire about certain inconsistencies in the publication.

Firstly, concerning the top half of Table 3 and the corresponding Figure 1, the authors state in the Table 3 description that “Group contrast terms are orthogonal, and effects were estimated from models holding constant medical comorbidities, HIV disease/treatment characteristics, and other lifetime substance use. β estimates are equivalent to the difference in T scores between groups (e.g., compared to M+C+, M+C− displayed [β = −2.09] lower T scores)” [page 8]. Upon inspection of the corresponding Figure 1, there seem to be inconsistencies between the data visualization and the reported values in Table 3. The authors state that Figure 1 is a “Profile plot of domain T score–predicted values in 472 PLWH from generalized linear regression models, controlled for medical comorbidities, HIV disease/treatment characteristics, current depressive symptoms, estimated premorbid verbal IQ, and other lifetime substance use. Groups represent lifetime DSM-IV substance abuse/dependence diagnoses for cannabis (C+/C−) and methamphetamine (M+/M−)” [page 9]. If this is true, then the reader should be able to subtract any two domain T-Score values between substance use groups and obtain the reported β values in the table. However, this does not seem to be true. For example, in Table 3, the beta estimate for “Verbal” is reported as 0.84, meaning M+C− displays [β = 0.84] higher T scores compared to M+C+. However, in Figure 1, it is clearly seen that M+C+ has a higher Verbal T-score compared to M+C−. Similar issues are found in other column comparisons. Either the data visualization is incorrect, the reported values are incorrect, or perhaps the data visualization is based on the raw data, in which case the label and description would be incorrect.

Furthermore, concerning the bottom half of Table 3 and the corresponding Figure 2, the authors state in the Table 3 description that “OR represents the odds ratio, or comparative difference in odds of displaying domain impairment (e.g., compared with M+C+, M+C− displayed 61% greater odds of global impairment)” [page 8]. Upon inspection of Figure 2, which the authors describe as a “Profile plot of predicted probability of domain impairments in PLWH (*n* = 472) from binomial regression models, controlled for medical comorbidities, HIV disease/treatment characteristics, current depressive symptoms, estimated premorbid verbal IQ, and other lifetime substance use. Groups represent lifetime DSM-IV substance abuse/dependence diagnoses for cannabis (C+/C−) and methamphetamine (M+/M−)”. [page 10], it seems that the data visualization of the probabilities of impairment in the substance use groups (Figure 2) does not correspond with the authors’ reported odds ratios of impairment between the substance use groups (Table 3). The reader should be able to calculate these odds ratios reported in Table 3 from Figure 2. However, they do not align. Again, we thought that perhaps the data visualization in Figure 2 was taken from the raw data, in which case the description of the figure would be incorrect.

In conclusion, due to the importance of this research and its implications for the field, we find it imperative to point out these inconsistencies in this paper. The points we have mentioned may not encompass all irregularities, so we highly recommend further peer review. Thank you for taking the time to read our correspondence, and we look forward to hearing from you soon. 

## References

[B1-viruses-15-01753] Rogers J.M., Iudicello J.E., Marcondes M.C.G., Morgan E.E., Cherner M., Ellis R.J., Letendre S.L., Heaton R.K., Grant I. (2023). The Combined Effects of Cannabis, Methamphetamine, and HIV on Neurocognition. Viruses.

