# Peer review of "Comment on Rogers et al. The Combined Effects of Cannabis, Methamphetamine, and HIV on Neurocognition. Viruses 2023, 15, 674"

_viruses, 2023, doi:10.3390/v15081753_

Round 1

Reviewer 1 Report

In this letter to the editor, the authors pointed out some discrepancies in the data analysis of a previously published paper (Rogers et al Viruses 2023 Mar; 15(3): 674) in this journal. The main concerns were in the data analysis related to Figure 1, Figure 2 and Table 3 of the manuscript. I agree with the authors concerns presented here related to the data analysis of the Rogers et al manuscript. Since this topic is highly relevant to multiple fields such as HIV, substance abuse and neurocognition and brain function, highly recommend this letter to be published and necessary discussion with the authors of the Rogers et al to address the concerns raised in this correspondence.

Quality of the English is acceptable..

Author Response

Thank you for the feedback.  I made one minor punctuation edit in line 51.

Reviewer 2 Report

This commentary was very well organized and to the point. The authors make an important point and provide concrete reasons for sending the publication in question back out for review or at least back to the original authors for clarification. It is worthy of publication with one minor edit on line 51 where there should be a period before 'however' rather than a comma.

As noted, just one edit is recommended but it is otherwise well-written.

Author Response

Thank you for the feedback.  I have corrected the grammatical error on line 51.